# Undernutrition Scored Using the CONUT Score with Hypoglycemic Status in ICU-Admitted Elderly Patients with Sepsis Shows Increased ICU Mortality

**DOI:** 10.3390/diagnostics13040762

**Published:** 2023-02-17

**Authors:** Junko Yamaguchi, Kosaku Kinoshita, Katsuhiro Nakagawa, Minori Mizuochi

**Affiliations:** Division of Emergency and Critical Care Medicine, Department of Acute Medicine, Nihon University School of Medicine, 30-1 Oyaguchi Kamimachi, Itabashi-ku, Tokyo 173-8610, Japan

**Keywords:** undernutrition, hypoglycemia, sepsis, intensive care

## Abstract

This study aimed to clarify whether the influence of undernutrition status and the degree of glycemic disorders affected the prognosis of patients with sepsis. A total of 307 adult patients with sepsis were retrospectively enrolled and analyzed. Characteristics, including nutrition status, calculated according to the Controlling Nutritional Status (CONUT) score of survivors and non-survivors, were examined. The independent prognostic factors of these patients with sepsis were extracted using multivariable logistic regression analysis. The CONUT scores in three glycemic categories were compared. Most patients with sepsis (94.8%) in the study had an undernutrition status according to their CONUT scores. High CONUT scores (odds ratio, 1.214; *p* = 0.002), indicating a poor nutritional status, were associated with high mortality. The CONUT scores in the hypoglycemic group were significantly higher than those in other groups with an undernutrition status (vs. hyperglycemic, *p* < 0.001; vs. intermediate glycemic, *p* = 0.006). The undernutrition statuses of patients with sepsis in the study scored using the CONUT were independent predictors of prognostic factors.

## 1. Introduction

Malnutrition is closely related to poorer clinical outcomes in the intensive care unit (ICU) [1].

The incidence of undernutrition in the ICU is high. Patients with sepsis and undernutrition may have a prolonged recovery duration in the ICU, which relates to a poor prognosis [2]. However, the reported incidence of undernutrition in ICUs varies widely and results for increased mortality vary in terms of whether the difference is statistically significant. The prevalence of undernutrition in the ICU ranges from 37.8% to 78.1%, which is higher odds (OR, 1.6; *p* = 0.02) of undernutrition compared to those in the general ward.

This appears to be due to variations in the accuracy and methods of assessment of undernutrition in ICUs and because studies include a wide range of background diseases [1]. The effects of sepsis on the prognosis of patients with undernutrition before they become septic have not been clearly determined.

In addition, a glycemic disorder in sepsis is a well-known prognostic factor in patients with sepsis, mainly due to hyperglycemia caused by insulin resistance, but hypoglycemia has been reported in a small number of critically ill patients. Regarding hypoglycemia and outcomes in the critically ill, patients with severe hypoglycemia show nearly two times higher fatality rates than those without hypoglycemia [3,4,5,6]. We also previously reported that mortality was increased approximately five times higher in septic patients with combined hypoalbuminemia and hypoglycemia. Several possible mechanisms may lead to hypoglycemia in septic patients [7]. The mechanism that is thought to produce hypoglycemia may be influenced by background undernutrition. Several undernutrition risk scoring systems exist for critically ill patients [8,9]. However, these parameters are potentially difficult to obtain from critically ill patients [10].

In the CONUT system, only serum albumin levels, total lymphocyte counts, and cholesterol levels in the peripheral blood are included in the nutritional status assessment [11]. However, albumin levels in acute conditions, including sepsis, decrease in response to various inflammatory effects, such as vascular permeability, dilution, Zn deficiency, and others. It is not necessarily a factor that indicates undernutrition [12]. Besides, in recent years, various complete blood count (CBC) markers have been used as indicators of inflammation markers (ratios of lymphocytes, leukocytes, and platelets). Among such markers, NLR (neutrophil/lymphocyte ratio) and PLR (platelet/lymphocyte ratio) are increasingly seen as alternative indicators of nutritional status [13].

Conversely, it has long been known that hypocholesterolemia is found in patients with sepsis, and the association between its degree and prognosis has been suggested [14]. Although data on the cause of reduced cholesterol levels in basic research are scarce, our previous study found that reduced lecithin-cholesterol acyltransferase (LCAT) activity due to increased oxidative stress is a cause of reduced cholesterol levels [15]. Further-more, albumin is a transport carrier of cholesterol and other molecules, likely affecting the fluctuation in fatty acids and outcomes in sepsis [16]. Cholesterol, which maintains the normal activity of the cell membrane, is affected by albumin, and both are thought to exacerbate sepsis conditions. Based on these findings, we thought it would be appropriate to examine nutritional assessment and sepsis outcomes using the CONUT score, which includes cholesterol levels, rather than albumin carriers. This study aimed to clarify whether undernutrition status is scored by a CONUT score and whether the degree of glycemic disorders affects the poor ICU mortality of patients with sepsis.

## 2. Materials and Methods

This study was a single-institution observational investigation that used the database of patients treated for sepsis at our hospital and was approved by the Clinical Research Review Committee of the Nihon University School of Medicine (RK-2011011). The need for informed consent for the study was waived by the approving authority.

The enrolled patients were classified according to the Third International Consensus Definitions for Sepsis and Septic Shock, defined as infections with systemic manifestations [17]. They were admitted to the ICU of this hospital between January 2016 and December 2019. All data for this study were obtained from the database and clinical records of the patients. Patients who were already diagnosed with sepsis and received treatment at other hospitals before admission to our hospital, as well as cases where details regarding the pre-hospital events were incomplete, were excluded from this study.

Blood cultures and whole-body CT scans of the infection sources were performed to confirm the diagnosis of infectious illness. Peripheral whole blood was collected from the patients at admission. Patient information and laboratory data were recorded, including age, sex, Acute Physiology and Chronic Health Evaluation (APACHE) II scores [18], and the Sequential Organ Failure Assessment (SOFA) scores [19]. The body mass index (BMI) and Controlling Nutritional Status (CONUT) scores were recorded upon admission to assess the patients’ nutritional status at baseline [11,20].

Cases were classified into three groups according to the blood glucose levels (BGL) measured at admission: a hypoglycemia (Hypo-G) group (BGL < 80 mg/dL), a hyperglycemia (Hyper-G) group (BGL ≥ 200 mg/dL), and an intermediate glycemia (Inter-G) group (BGL 80–199 mg/dl) [20,21]. In this study, a blood albumin level < 2.8 mg/dL was defined as hypoalbuminemia (Hypo-A) and is associated with mortality according to the receiver operating characteristic (ROC) curve-derived cut-off values (area under ROC, 0.68; sensitivity, 0.72; 1-specificity, 0.43).

BGL in patients with hypoalbuminemia at the time of admission was divided into separate groups and analyzed according to the method used in a previous study [7]. Hemoglobin A1c (HbA1c), per the National Glycohemoglobin Standardization Program, was also included as a chronic glycemic status indicator. Patients in the study were categorized as having diabetes or without diabetes using HbA1c levels > 6.5% as cut-off values for evaluating their septic severity, nutrition status, and prognosis. Outcomes were evaluated at the time of discharge from the ICU.

### 2.1. CONUT Score

Undernutrition is a significant problem in clinical situations; therefore, early detection is important. Ideally, a screening tool for undernutrition should be clear, accurate, easy to use, and cheap. Several undernutrition risk scoring systems for critically ill patients, such as the Subjective Global Assessment (SGA) and the Nutritional Risk Screening 2002, have been globally used [8]. Weight loss, oral intake in the prior week, and the number of comorbidities were included as parameters [9]. However, these parameters are potentially difficult to obtain from critically ill patients [10]. In 2019, the Global Leadership Initiative on Malnutrition (GLIM) was proposed as an index of malnutrition [8], and it is being used as an undernutrition evaluation index for acute inflammatory conditions. However, it has not been widely used in ICU settings. When using the CONUT system, only serum albumin levels, total lymphocyte counts, and total cholesterol levels in the peripheral blood are included in the assessment of the nutritional status. The method of assessing the undernutrition status using the CONUT scoring system is shown in Appendix A. The CONUT score has been previously validated with results correlating with those of the SGA [11]. In addition, new evidence has shown a better correlation between GLIM and CONUT scores in acute patients [22]. The CONUT score is thus an efficient, accurate, and easy-to-use tool for the early detection and continuous control of hospital undernutrition.

### 2.2. Statistical Analyses

All the analyses were performed using SPSS (version 25; IBM Corp., Armonk, NY, USA) and JMP ver. 14.2 (SAS Institute, Cary, NC, USA). The data were presented as mean values (standard deviation (SD)) or the number of cases (%). The statistical significance was set at *p* < 0.05. The continuous variables were compared using the Student’s t-test or Mann–Whitney U test, as appropriate. The chi-square or Fisher’s exact probability tests was performed for the categorical variables. The physiological data from each glycemic condition group were compared using one-way analyses of variance or a Kruskal–Wallis analysis. Subsequently, Steel–Dwass post hoc tests were performed.

The outcomes were predicted using multiple logistic regression analysis, and the odds ratios (ORs) and 95% confidence intervals (CIs) were calculated. The variables with *p*-values < 0.2 obtained from the bivariate analysis were introduced into the multivariate models [23]. Finally, Kruskal–Wallis or Steel–Dwass tests were performed to compare the CONUT scores in the three glycemic categories. Multivariate models were used to determine the previously described clinical factors (APACHE II and SOFA score systems) related to the outcomes as explanatory variables.

Both systems have already been used globally as indicators for the clarification of disease severity, which has already been validated. Table 1 shows many explanatory variables, including the APACHE II and SOFA scores, platelet count, and albumin, bilirubin, and creatinine levels. However, platelet count and bilirubin and creatinine levels are also included in the APACHE II and SOFA score systems, and albumin level is also included in the CONUT score. We were concerned with influences of multicollinearity occurring if we included all factors. Therefore, we excluded duplicated factors to avoid multicollinearity. In addition, all variables with *p*-values of <0.2 in the bivariate models were transferred to the multivariate models (multiple logistic regression analysis). Multicollinearity, which was assessed using the variance inflation factors, was detected among age [24], platelet count and bilirubin and creatinine levels, APACHE II scores, SOFA scores, CONUT scores, BGL (for each glycemic group), and blood albumin levels (in cases with hypoalbuminemia), as per the appropriate selection in the multivariate models.

## 3. Results

A total of 556 patients with sepsis were enrolled during the study period. After excluding 249 who had already begun treatment at another hospital, those with incomplete data, and those aged < 19 years, 307 patients with sepsis (186 men and 121 women) were included in this study. The origin of the infections and causative pathogens among groups are shown in Figure 1 and the origins of infection in each glycemic group are shown in Appendix A. Figure 2 and Table 2 show comparisons of the origin of infection. The ratio of survivors was significantly higher than that of non-survivors in the urinary tract group and various other organ groups. Moreover, the ratio of non-survivors was significantly higher than that of survivors in the endocardial system (Figure 2). No significant differences were observed in the CONUT score, BG level, and SOFA score. For glycemic conditions, the ratio of Hypo-G significantly differed according to the origin of the infection (Table 2).

The ratio of survivors showed no significant differences among the causative pathogen groups (Appendix A) and there were no significant differences in the SOFA score and origin of infection among the causative pathogen groups. The ratio of septic shock and the three glycemic groups showed no statistically significant differences by causative pathogens (Appendix A). Conversely, the CONUT score was significantly different among the causative pathogen groups: the score for the Gram-positive bacteria group was significantly higher than that for the none and the non-specific groups (Figure 3).

The patient groups were divided into survivors and non-survivors, and their characteristics and outcomes were compared (Table 1). In the non-survivor group, significantly higher values were obtained in the APACHE II/SOFA scores, higher total bilirubin, AST, creatinine, and lactate levels, lower platelet counts, blood albumin levels, and antithrombin III levels, and existing septic shock and metabolic acidosis states. However, age, sex, white blood cell counts, hemoglobin, hematocrit, serum sodium, serum potassium, blood urea nitrogen, C-reactive protein, and uric acid levels between the groups were not significantly different (Table 1A).

In terms of nutritional status, only 5.2% of the patients were normal, according to the CONUT scoring system. Almost all study patients had a moderate to severely impaired nutritional status. The CONUT score was significantly higher in the non-survivor group than that in the survivor group. In particular, the number of patients in the severe category assessed by the CONUT scoring system was significantly higher in the non-survivor group than in the survivor group (50.0% in non-survivors vs. 30.4% in survivors, *p* = 0.004). However, the BMI between the two groups was not significantly different (Table 1B). The patients included in the study were divided into three groups based on their BGL at admission. The number of patients in each group and the distribution of all the glycemic groups are shown in Table 1C. While there was a significant difference between survivors and non-survivors, there was no significant difference in the number of patients with Hypo-A between the two groups (Table 1C). In addition, BGL was significantly higher in the survivor group, while HbA1c levels between the groups did not show a significant difference (Table 1D). The multiple logistic regression analyses of the initial laboratory data at admission showed that high SOFA scores (OR 1.178; 95% CI 1.048–1.323; *p* = 0.006), a higher lactate level (OR 1.192; 95% CI 1.100–1.290; *p* < 0.0001), or high CONUT scores (OR 1.214; 95% CI 1.076–1.368; *p* = 0.002), indicating poor nutritional status, were associated with higher mortality. The other independent predictors of a high mortality risk did not demonstrate a correlation (Table 3).

The CONUT scores of each glycemic group are shown in Figure 4. The CONUT scores showed a statistical difference among the three glycemic groups (Hyper-G, 5.8 ± 0.38; Inter-G, 6.9 ± 0.26; Hypo-G, 8.7 ± 0.46; *p* < 0.0001). However, the CONUT scores in the Hypo-G group were the highest and were significantly higher than those in the other two groups (Hyper-G, *p* < 0.001; Inter-G, *p* = 0.006, respectively).

The comparison of baseline characteristics between patients with and without diabetes indicates chronic glycemic status, as shown in Table 4. There were no statistically significant differences in their outcomes and general nutritional statuses, such as BMI and CONUT score, except for the total cholesterol level, which is one of the components of the CONUT score. BGL and HbA1c values in the diabetes group were significantly higher than in the non-diabetes group. The ratio of glycemic abnormality groups, such as Hyper-G and Hypo-G, was shown in the diabetes group to have significant differences. Total cholesterol level in the Hyper-G group was higher than that of the non-Hyper G group (169.6 ± 62.6 vs. 144.1 ± 72.0, respectively; data not shown).

## 4. Discussion

This study showed a statistically significant difference in the number of cases with glycemic abnormalities between the groups. The number of hypoglycemia cases in the non-survivor group was significantly higher than that in the survivor group. The undernutrition scores, according to the CONUT scores in the hypoglycemia group in our study, were significantly higher than those in the other two glycemic groups.

The multiple consequences of undernutrition affecting many organ systems are well-known. They are associated with several other complications, as well as extensions in the length of hospital stays, causing increases in hospital assistance costs [11]. Giner et al. reported that 40% of ICU patients were malnourished, and this was related to poor outcomes [25].

The mortality rate of patients with sepsis continues to be high. Thus, prompt clinical evaluation and prognosticating morbidity and mortality are required during intensive care. The severity and prognosis of these patients have been generally evaluated using the APACHE II and SOFA scores [18,19,26]. Patients with sepsis with undernutrition may also have prolonged recovery durations while in the ICU, which is related to an undesirable prognosis [2]. Recently, Godinez-Vidal et al. reported that the malnutrition determined by the CONUT was related to severity and mortality in patients with abdominal sepsis [27].

This study’s results indicated that the undernutrition status, determined by the CONUT score, of patients with an infection source other than abdominal sepsis was also an independent predictor of high mortality risk. Therefore, the CONUT score may be used not only for assessing the nutritional status, but also for assessing the severity and short-term prognoses of patients with sepsis.

Previous reports demonstrated that hypoglycemia was also a prognostic factor in patients with sepsis [3,4,5,6]. The pathophysiology of hypoglycemia in sepsis may involve many mechanisms [28,29,30]. Hypoalbuminemia is known as a predictor of mortality [28,29]. As reported in our previous study, hypoglycemia with hypoalbuminemia was associated with a higher ICU mortality in patients with sepsis [7]. Undernutrition may not only lead to hypoalbuminemia but may also accelerate glycemic abnormalities, especially hypoglycemia. The results of this study may indicate that depending on the severity of the sepsis inflammation, the undernutrition status accelerated undesirable prognoses in patients with sepsis, leading to concomitant hypoglycemia, although it may have increased albumin synthesis and transcapillary loss, because of inflammatory reactions. The underlying mechanisms regarding how hypoglycemia in patients with sepsis and undernutrition conditions are related to outcomes remain unclear. It is thus important to clarify, using prospective observational studies, the mechanisms by which glucose metabolism in patients with sepsis is affected. We would like to add our considerations on changes in sepsis other than albumin. This study used the CONUT score because its items included cholesterol levels. It has long been known that hypocholesterolemia is found in patients with sepsis, and an association between its degree and prognosis has been suggested. A decline in the fat absorption function of the intestinal tract, decreased cholesterol synthesis, cholesterol transport disorder, and abnormal metabolic function due to toxin removal in sepsis have been suggested as the causes [14]. In our previous study, we found that reduced LCAT activity due to increased oxidative stress is a cause of reduced cholesterol levels [15], and albumin is a transport carrier of cholesterol and other molecules, likely affecting the fluctuation in fatty acid and outcomes in sepsis [16].

Cholesterol, which maintains the normal activity of the cell membrane, is affected by albumin, and both are thought to exacerbate sepsis conditions. Based on these, we thought it would be appropriate to examine nutritional assessment and sepsis outcomes using the CONUT score, which includes cholesterol levels, rather than albumin carriers. According to a review on the role of cholesterol in sepsis [14], cholesterol is also involved in immunomodulation and antibacterial effects, and hypocholesterolemia has an adverse effect on natural and acquired immunity. In addition, it has been suggested to decrease the production of adrenal steroids, sex hormones, and vitamin D, affecting their deficiency. Adrenal gland dysfunction is known as a complication of sepsis and septic shock. However, many patients who die from sepsis have increased cortisol levels in plasma but a reduced response to ACTH stimulation [13]. Thus, cholesterol has been suggested to induce patients into a hypoglycemic condition.

While a glycemic disorder was observed in non-survivors in our study, glycemic abnormalities in each patient with sepsis using the measured HbA1c levels were already determined as unrelated to mortality. Stengenga et al. reported that the relationship between hyperglycemic condition at admission and poor prognosis was observed only without patients with diabetes and sepsis [31]. The results of our study supported the findings of other studies that hyperglycemia in admitted patients with sepsis carried a high mortality risk; however, diabetes was not a mortality risk [31,32,33]. Acute stress insults produce acute hyperglycemia induced by the hypothalamic–pituitary–adrenal axis and the sympathetic nerve reactions. These hyperglycemic conditions lead to endothelial dysfunction, cytokine release, mitochondrial dysfunction, and accelerated coagulation systems, leading to multiple organ failures in patients with sepsis [33]. However, long-term hyperglycemic conditions may be protective, as observed in the protective downregulation of glucose transporter (GLUT)-4 and GLUT-1 expressions [33]. Differences in pathophysiological mechanisms and roles, such as pre-existing diabetes, might exist between acute hyperglycemia and chronic hyperglycemia.

This study had several limitations. A major limitation was that this was a retrospective observational study with a limited number of patients in a single institution. Moreover, the outcomes of patients with sepsis are affected by their comorbidities and pathogens that affect metabolic dynamics. Regarding comorbidities, there were many different comorbidities in our patient sample, and we did not have complete data regarding these comorbidities. In addition, identifying the causative pathogen was often difficult in the clinical setting of the study. The non-specific group included different patients, which could have resulted in bias. Although BMI was not related to outcomes in patients with sepsis in our study, almost all the patients included were older patients, with an average age of 74.7 years. For estimating influences in the study due to adding age, in the analysis of comparing patients of <65 years of age, there were not any statistical differences in sepsis severity and nutrition status scored by the CONUT score and glycemic condition (table was not shown). However, there was an average BMI of 20.7 in our study. Since the results of this study differed from those of previous reports, care should be taken with their interpretation. However, with the increased number of older patients, sarcopenia induced by age has increased in all developed countries. Therefore, our results might apply to future patients with sepsis. Moreover, the patients’ nutritional status was not evaluated clearly and correctly using other classical nutritional evaluation methods such as the SGA, and the duration of the illnesses before hospitalization could not be evaluated. Hypoalbuminemia is known as a predictor of mortality by itself [28,29], and the albumin level in acute conditions, including sepsis, decreases in response to various inflammatory effects. There is a plethora of evidence suggesting that albumin levels are not always dependent on nutritional status [12]. However, we believe that incorporating total cholesterol levels into the CONUT score will facilitate the assessment of patients with mild malnutrition who do not present with decreased albumin levels. With a half-life of 8 days, total cholesterol is not as sensitive as rapid turnover proteins such as albumin, which has a half-life of 17–23 days. Furthermore, while histories of diabetes were evaluated using the HbA1c values on admission, accurate data regarding nutritional intake or use of antidiabetic medicine in patients with a history of diabetes could not be obtained. Generally, estimating the duration of illness is difficult because many patients with sepsis do not know the actual time of onset. For these reasons, this study could only identify patients with long durations of illnesses, as reflected in the collapsed pathophysiological conditions after systemic depletion due to infections. The data were only obtained from the patients at the time of admission. Thus, the calculation from the ROC curves for the CONUT score and outcome in this study is not precise. The highest AUROC was observed for a CONUT score > 10 points (0.66; 95% CI 0.58–0.75), and there is weak evidence to support that the CONUT score was an independent prognostic factor in septic patients (the figure is not shown). Previous studies have shown that hypoglycemia is a prognostic factor in septic patients, and the present study indicates that malnutrition as indicated by the CONUT score predicts a severe prognosis in septic patients. However, it is unclear why the hypoglycemia group had a more common prevalence of malnourished patients with high CONUT scores.

The results of this study suggest that the low nutritional status indicated by the CONUT score might be a predictor of the outcome in patients with sepsis because it indirectly alters glucose metabolism and cholesterol, involved in various functions, such as affecting immune responses. The present results suggest that there is room for further investigation. In the future, it would be desirable to examine the correlation between the progression of sepsis and the CONUT score after ICU admission, including measurement of cortisol and other parameters to elucidate the relationship between nutritional status and outcome more accurately in a prospective study.

In summary, the undernutrition statuses of patients with sepsis on admission to the study scored using the CONUT were independent predictors of prognostic factors. The CONUT score is useful not only for assessing the nutritional status but also for assessing the severity and short-term prognosis of patients with sepsis. Glycemic disorders in patients with sepsis, especially hypoglycemia, were related to undernutrition on admission, as determined by the CONUT score. However, previous glycemic conditions indicated by HbA1c values were not related to any prognostic factor.

## Figures and Tables

**Figure 1 diagnostics-13-00762-f001:**
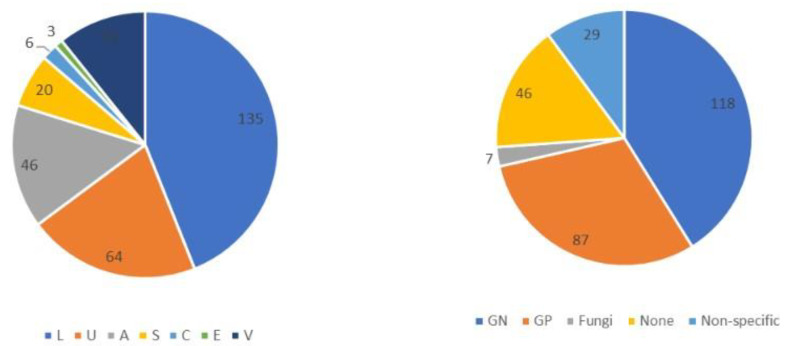
Origin of infection and causative pathogens among groups in this study (*n* = 307). Abbreviations: L, lung; U, urinary tract; A, abdomen; S, soft tissue; C, central nervous system; E, endocardial systems; V, various other organs, including multiple infectious origins and cases of unknown origin. GN, Gram-negative bacteria; GP, Gram-positive bacteria. Non-specific, no causative pathogenic bacteria were identified, normal flora or multiple bacteria were detected but could not be found as causative pathogenic bacteria.

**Figure 2 diagnostics-13-00762-f002:**
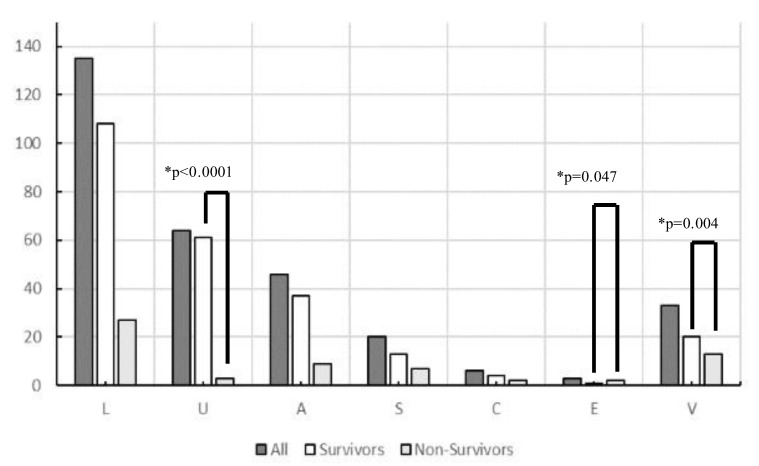
Origin of infection in survivors and non-survivors in this study (*n* = 307). Abbreviations: L, lung; U, urinary tract; A, abdomen; S, soft tissue; C, central nervous system; E, endocardial systems; V, various other organs, including multiple infectious origins and cases of unknown origin. * Chi-square tests were performed to analyze the ratio of survivors and non-survivors.

**Figure 3 diagnostics-13-00762-f003:**
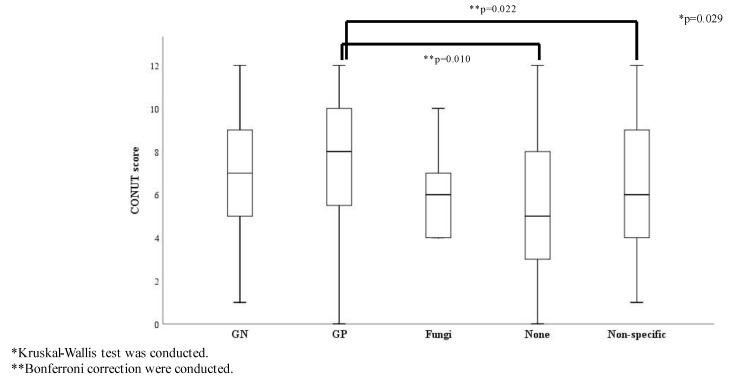
CONUT score comparisons according to causative pathogens (*n* = 307). Abbreviations: GN, Gram-negative bacteria; GP, Gram-positive bacteria; Non-specific, normal flora or multiple bacteria were detected but causative pathogenic bacteria could not be identified.

**Figure 4 diagnostics-13-00762-f004:**
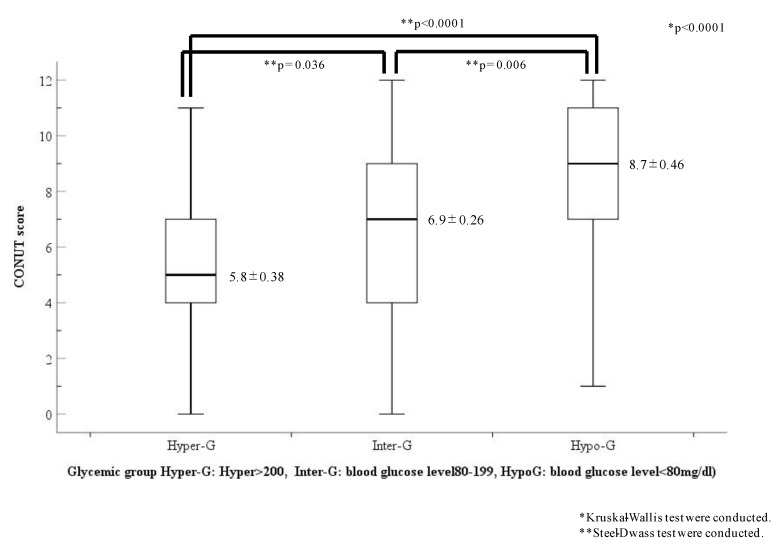
Comparison of CONUT scores in the three glycemic categories. Abbreviations: Hypo-G, hypoglycemia group (blood glucose level < 80 mg/dL); Hyper-G, hyperglycemia group (blood glucose level ≥ 200 mg/dL); Inter-G, intermediate glycemia group (blood glucose level = 80–119 mg/dL).

**Table 1 diagnostics-13-00762-t001:** Comparison of survivors and non-survivors.

	All (*n* = 307)	Survivors (*n* = 244)	Non-Survivors (*n*= 63)	*p*-Value *
(A) Parameters				
Age	74.7 ± 14.4 [21–99]	74.7 ± 14.1 [21–99]	74.9 ± 15.7 [24–98]	0.547
Sex (M:F)	121:186	149:95	37:26	0.735
APACHE II score	24.7 ± 7.52 [2–43]	23.9 ± 7.47	27.5 ± 7.06	0.001
Ratio of septic shock (%)	152/301 (50.5%)	103/240 (42.9%)	49/61 (80.3%)	<0.0001
SOFA score	8.1 ± 3.3 [0–19]	7.6 ± 3.1	10.0 ± 3.3	<0.0001
WBC (×10^3^/µL)	12.8 ± 8.17	13.0 ± 8.1	12.2 ± 8.3	0.503
Hemoglobin (g/dL)	11.9 ± 3.0	11.9 ± 2.9	12.0 ± 3.1	0.802
Hematocrit (%)	37.2 ± 23.1	37.5 ± 25.4	36.5 ± 9.12	0.848
Platelet (×10^4^/µL)	196.5 ± 112	205 ± 112	165 ± 107	0.002
Albumin (g/dL)	2.82 ± 0.75	2.91 ± 0.74	2.48 ± 0.72	<0.0001
T. bilirubin (mg/dL)	1.19 ± 1.92	0.97 ± 0.91	2.03 ± 3.74	0.003
AST	144 ± 404	134 ± 420	183 ± 332	0.002
ALT	85 ± 331	83 ± 354	92 ± 221	0.165
Na	139 ± 11	140 ± 8.51	136 ± 17.9	0.101
K	4.47 ± 1.04	4.46 ± 1.03	4.49 ± 1.09	0.546
BUN	51.4 ± 42.6	50.3 ± 43.2	55.5 ± 40.4	0.091
Creatinine (mg/dL)	2.18 ± 2.50	2.13 ± 2.66	2.37 ± 1.76	0.011
Lactate (mmol/L)	4.92 ± 4.4	4.10 ± 3.50	8.05 ± 5.87	<0.0001
HCO3- (mmol/L)	19.2 ± 7.21	20.1 ± 7.01	15.9 ± 7.05	<0.0001
CRP (mg/dL)	14.3 ± 12.7	13.8 ± 12.6	16.4 ± 12.8	0.108
AT3	74.0 ± 21.5	76.7 ± 16.9	63.7 ± 31.9	<0.0001
UA	7.9 ± 4.2	7.80 ± 4.23	8.27 ± 4.17	0.273
(B) Nutritional status				
BMI	20.7 ± 7.8	20.6 ± 4.8	19.7 ± 4.3	0.198
CONUT score	6.9 ± 3.2	6.5 ± 3.1	8.3 ± 3.0	<0.0001
Lymphocyte count	92 ± 1021 [30–9120]	950 ± 1037	796 ± 957	0.061
T-Cho (mg/dL)	150.1 ± 70.5 [38–853]	150.9 ± 54.7	147.3 ± 114	0.026
Normal category	13/252 (5.2%)	12/204 (5.9%)	1/48 (2.1%)	0.253
Light category	56/252 (22.2%)	49/204 (24.0%)	7/48 (14.6%)	0.197
Moderate category	97252 (38.5%)	81/204 (39.7%)	16/48 (33.3%)	0.45
Severe category	86/252 (34.1%)	62/204 (30.4%)	24/48 (50.0%)	0.004
(C) Glycemic groups				
Hypo-G	46/304 (15.1%)	26/243 (10.7%)	20/61 (31.7%)	<0.0001
Inter-G	183/304 (60.2%)	156/243 (64.2%)	27/61 (44.3%)	<0.0001
Hyper-G	75/304 (24.7%)	61/243 (25.1%)	14/61 (23.0%)	<0.0001
Hypo-A	143/305 (53.1%)	105/242 (43.4%)	38/63 (60.3%)	0.303
(D) Glycemic status				
BG (mg/dL)	176 ± 161	183 ± 158	147 ± 168.3	0.003
HbA1c (NGSP (%))	6.1 ± 1.21	6.11 ± 1.13	5.98 ± 1.48	0.117

* Continuous variables were compared using Student’s t-tests or the Mann–Whitney U tests, as appropriate. Chi-square tests or Fisher’s exact probability tests were performed for categorical variables. We determined the optimal cut-off and the significance level at 5%. Abbreviations: APACHE II, Acute Physiology and Chronic Health Evaluation II; SOFA score, Sequential Organ Failure Assessment score; WBC, white blood cells; T. bilirubin, total bilirubin; AST, aspartate aminotransferase; ALT, alanine aminotransferase; CRP, C-reactive protein; AT3, antithrombin 3; UA, uric acid; BMI, body mass index; CONUT score, Controlling Nutritional Status score; BG, Blood Glucose; HbA1c, hemoglobin A1c; NGSP National Glycohemoglobin Standardization Program; Hypo-G, hypoglycemia group (blood glucose level < 80 mg/dL); Hyper-G, hyperglycemia group (blood glucose level ≥ 200 mg/dL); Inter-G, intermediate glycemia group (blood glucose level = 80–199 mg/dL); Hypo-A, hypoalbuminemia (blood albumin < 2.8 mg/dL).

**Table 2 diagnostics-13-00762-t002:** Origin of infection in this study.

Origin of Infection	Lung (*n* = 135)	Urine Tract (*n* = 64)	Abdomen (*n* = 46)	Soft Tissue (*n* = 20)	Central Nervous System (*n* = 6)	Endocardial Systems (*n* = 3)	Various Other Organs (*n* = 33)	*p*-Value
CONUT score *	6.52 ± 3.13	7.27 ± 3.46	7.36 ± 2.91	7.5 ± 3.14	6.83 ± 2.99	5.67 ± 2.52	6.81 ± 3.45	0.642
BG (mg/mL) *	168.4 ± 131.4	174 ± 141.5	185.2 ± 152.5	116.5 ± 70.9	215.7 ± 80.3	183.7 ± 42.3	223.5 ± 310.8	0.124
Ratio of septic shock (%) **	56/132 (42.4)	35/64 (54.7)	27/45 (60.0)	11/19 (57.9)	3/6 (50.0)	2/3 (66.7)	18/32 (56.3)	0.361
Ratio of Hypo-G (%) **	13/135 (9.6)	9/64 (14.1)	9/46 (19.6)	7/19 (36.8)	0/6 (0.0)	0/3 (0.0)	7/19 (36.8)	0.022
Ratio of Inter-G(%) **	91/135 (67.4)	40/64 (62.5)	22/46 (47.8)	10/19 (52.6)	4/6 (66.7)	2/3 (66.7)	14/31 (45.2)	0.145
Ratio of Hyper-G (%) **	31/135 (23.0)	15/64 (23.4)	15/46 (32.6)	2/19 (10.5)	2/6 (33.3)	1/3 (33.3)	9/31 (29.0)	0.603
SOFA score *	7.47 ± 2.80	8.17 ± 3.61	8.76 ± 3.70	8.45 ± 3.03	9.33 ± 3.39	8.00 ± 9.464	9.24 ± 3.48	0.074

Abbreviations: Various other organs, including multiple infectious origins and cases of unknown origin. CONUT score, Controlling Nutritional Status score; BG, blood glucose; Ratio of septic shock, ratio of septic shock/total number of patients; Hypo-G, hypoglycemia group (blood glucose level < 80 mg/dL); Hyper-G, hyperglycemia group (blood glucose level ≥ 200 mg/dL); Inter-G, intermediate glycemia group (blood glucose level = 80–119 mg/dL); SOFA score, Sequential Organ Failure Assessment score. SOFA score was obtained at admission (mean ± SD). * Kruskal–Wallis test was conducted for multiple comparisons of CONUT scores, BG, and SOFA scores, and the origin of infection. No statistically significant differences were found (*p* = 0.074, *p* = 0.124, *p* = 0.074, respectively). ** Chi-square tests were performed to analyze the ratio of septic shock and each of the three glycemic groups (Hypo-G, Inter-G, Hyper-G). We determined the optimal cut-off and the significance level at 5%.

**Table 3 diagnostics-13-00762-t003:** Independent predictors of non-survival.

Explanatory Variable	Odds Ratio	95% CI	*p*-Value *
Parameters			
APACHE II score	-		
SOFA score	1.178	1.048–1.323	0.006
AST	-		
ALT	-		
BUN	-		
Lactate	1.192	1.100–1.290	<0.0001
CRP	-		
AT3			
Glucose	-		
Hypo-G	-		
Inter-G	-		
Hyper-G	-		
CONUT score	1.214	1.076–1.368	0.002
BMI	-		
HbA1c	-		

* Predictive factors were analyzed using multiple logistic regression using the forced entry method. The clinical factors related to prognosis were used as explanatory variables. All variables with *p*-values < 0.2 in the bivariate models were analyzed using the multivariate models (multiple logistic regression analyses). Abbreviations: CI, confidence interval; APACHE II, Acute Physiology and Chronic Health Evaluation II; SOFA score, Sequential Organ Failure Assessment score; AST, aspartate aminotransferase; ALT, alanine aminotransferase; BUN, blood urea nitrogen; CRP, C-reactive protein; AT3, antithrombin 3; Hypo-G, hypoglycemia group (blood glucose level < 80 mg/dL); Hyper-G, hyperglycemia group (blood glucose level ≥ 200 mg/dL); Inter-G, intermediate glycemia group (blood glucose level = 80–199 mg/dL); Hypo-A, hypoalbuminemia (blood albumin < 2.8 mg/dL); CONUT score, Controlling Nutritional Status score; BMI, body mass index; HbA1c, hemoglobin A1c.

**Table 4 diagnostics-13-00762-t004:** Comparison of baseline characteristics according to the presence of diabetes.

Variable	Non-Diabetes	Diabetes	*p*-Value
Age (years)	75.1 ± 15.3	73.1 ± 11.2	0.084
APACHE II score	24.7 ± 7.53	24.6 ± 7.56	0.94
SOFA score	8.21 ± 3.38	7.77 ± 2.89	0.603
CONUT score	7.0 ± 3.10	6.43 ± 3.46	0.262
Albumin (g/dL)	2.79 ± 0.74	2.96 ± 0.76	0.110
Lymphocyte count (per/µL)	91.4 ± 107.1	94.9 ± 84.0	0.441
T-Cho (mg/dL)	148.7 ± 71.6	158.0 ± 66.1	0.041
BG (mg/dL)	134.6 ± 67.6	318.4 ± 272.0	<0.0001
HbA1c (NGSP (%))	5.58 ± 0.51	7.79 ± 1.35	<0.0001
BMI	19.6 ± 4.37	22.9 ± 5.17	<0.0001
Mortality (D/All (%))	51/235	11/69	0.499

Abbreviations: APACHE II, Acute Physiology and Chronic Health Evaluation II; SOFA score, Sequential Organ Failure Assessment score; CONUT score, Controlling Nutritional Status score; T- Cho, total cholesterol; BG, blood glucose; HbA1c, hemoglobin A1c; NGSP, National Glycohemoglobin Standardization Program; BMI, body mass index; D, death.

## Data Availability

The data that support the findings of this study are available from the corresponding author, J.Y., upon reasonable request.

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
