# Peer review of "Undernutrition Scored Using the CONUT Score with Hypoglycemic Status in ICU-Admitted Elderly Patients with Sepsis Shows Increased ICU Mortality"

_diagnostics, 2023, doi:10.3390/diagnostics13040762_

Round 1

Reviewer 1 Report

The authors have done a commendable job in outlining their research design and presenting the observations from their studies. However, the following changes must be incorporated within the manuscript before the manuscript can be considered for publication:

1. The beginning of the introduction section can be better written by including relevant statistics or some data about the current state of knowledge and research and the gaps therein regarding the correlation between the undernutrition status and the degree of glycemic disorders affected the prognosis of patients with sepsis.

 2. Page 1 Lines 42-46 represent a very long convoluted sentence and is not clear to the readers. The authors must rewrite this as two separate shorter sentences to improve the clarity of this sentence.

3.The authors do outline their results, but it would be much better understood and represented in the form of pie charts and bar graphs visually depicting the data. This must be included in the manuscript. The statistically significant differences must also be represented graphically.

4.Did the authors investigate the correlation between the different stages of sepsis progression and CONUT scores?

5. What were the key differences in age groups and sex groups as classified in the study?

Reviewer 2 Report

Dear Authors,

I would like to congratulate you with results of the retrospective single center study presented in manuscript:” Undernutrition scored using the CONUT score with hypoglycemic status in ICU-admitted patients with sepsis leads to increased ICU mortality”.

The results are interesting, and the subject is novel.

First, I would like to suggest, if the title should not be changed as the analysis was performed on elderly patients, so maybe this information should be added into the title. 

Second, I haven’t seen any information about sepsis criteria. Would you be so kind to specify the definition of sepsis, that was the main indicator for patient’s enrollment.

You presented the CONUT score >1,214 as a higher mortality marker. Would you be so kind and present the ROC analysis? 

The results of multivariable analysis were significant for higher mortality rates in SOFA, CONUT and lactate levels. Why have you chosen of CONUT? 

Why you find so important to differentiate CONUT scores related to glycemic results? Don’t you find these two co-variates as too complicated to be useful in everyday practice?

From my humble opinion, the manuscript’s results are based on CONUT results subdivided by glycemia results that makes the manuscript not suitable for clinical application and though interesting, very unlikely to be repeatable in clinical practice.

Kinds 

Reviewer

Reviewer 3 Report

Authors presented a single-institution observational study investigating the influence of undernutrition status and the degree of glycemic disorders on the prognosis of patients with sepsis hospitalized at the ICU. The study is very interesting and indicates the role of Controlling Nutritional Status (CONUT) score in assessing the malnutrition among ICU patients and it's connection to survival prognosis.

However, the study needs extenisive English linguistic editing due to a large number of colloquialisms and stilistic mistakes.

Also some summarizing figure/table should be added.

IMHO the paper after corrections may be considered for publication.

Round 2

Reviewer 2 Report

Dear Authors,

I would like to thank you for your corrections in submitted manuscript titled "Undernutrition scored using the CONUT score with hypoglycemic status in ICU-admitted elderly patients with sepsis shows increased ICU mortality"

The manuscript was improved, and I’m impressed with the results.

Congratulations!

Kinds

Tom 

Reviewer 3 Report

The paper now may be considered for publication.